# Spatial variation and determinants of solid fuel use in Ethiopia; Mixed effect and spatial analysis using 2019 Ethiopia Mini Demographic and Health Survey dataset

Jember Azanaw[1]*, Gashaw Sisay Chanie[2]

1 Department of Environmental and Occupational Health and Safety, Institute of Public Health, College of Medicine and Health Sciences, University of Gondar, Gondar, Ethiopia, 2 Department of Clinical Pharmacy, School of Pharmacy, University of Gondar, Gondar, Ethiopia

* jemberazanaw21@gmail.com

## Abstract

**Data Availability Statement:** All relevant data are within the paper and its Supporting Information files.

### Background

According to UNSD, World Bank, and WHO, a 2020 joint report, 3 billion people who used solid fuel were found in low and middle-income countries. The burning of such type of fuel emits a variety of pollutants such as PM2.5, PM10, CO, NO2, organic compounds, and other substances which a had wide range of public health problems The evidence from the WHO report, indoor air pollution was attributed to around 3.2 million deaths each year in 2020, and more than 237 000 deaths of children under the age of five. This study aims to investigate solid fuel prevalence, predictors, and spatial variation in Ethiopia.

### Method

This study was conducted in Ethiopia based on the fifth Ethiopian Demographic and Health Surveys 2019 dataset. 8,663 households were successfully interviewed at a response rate of 99%. Weighted by sampling weight was done to do a reliable statistical analysis. Fuel type was used as the outcome variable whereas sex of household head (male or female), wealth index (poor, middle, and rich), educational status (no education, primary, secondary, higher), having television and radio, a separate room used as a kitchen, were individual predictor variables and community level education(lower/higher), residence(urban/rural), community level media exposure (exposed/unexposed), region (pastoralist, semi-pastoralist, Agrarian, and City administration) were community level variables. All the above analyses were conducted using Excel Microsoft 2016, STATA 14, ArcGIS 10.7, and SaTScan 10.1 software.

### Results

The majority (72.62%) of household heads were males. The overall national level of solid fuel use was 87.13% (95% CI (86.4%-87.82%)). From this 87.13% of solid fuel use, 18.60% accounts for urban and 68.53% for rural parts of the country. Educational status, having

**Funding:** The author(s) received no specific funding for this work.

**Competing interests:** The authors have declared that no competing interests exist.

**Abbreviations:** AIC, Akaike's Information Criterion; AOR, Adjusted Odds Ratio; CI, Confidence Interval; DHS, Demographic and Health Survey; EAs, Enumeration Areas; EDHS, Ethiopian Demographic and Health Survey; ICC, Intra Class Correlation; MOR, Median Odd Ratio; PCV, Proportional Change in Variance.

television, accessing electricity, and wealth index were individual-level variables community-level education, type of residence, and region were community-level variables significantly associated factors towards solid fuel use in our study. Global (z-score = 33.109144, P-value <0.0001), local (hot spot, cluster, and outlier), and Spatial Scan statistical analyses revealed that there was a major geographical variation in solid fuel use across Ethiopia.

## Conclusion

Based on this finding, the prevalence of solid fuel use was higher in Ethiopia. Educational status, having television, accessing electricity, and wealth index were individual-level variables and community-level education, type of residence, and region were community-level variables statistically significant factors in determining fuel choice for cooking. There was significant spatial variation in the prevalence of solid-fuel use across the country. In order to addressing such heavily dependent on solid fuel use, responsible bodies like health policy makers, national and international organizations, and public health researchers should work on showing health problems of solid fuel use and the means of increasing clean fuel use. Substantial policy modifications are desirable to reach access to clean fuels and technologies (SDG 7.1.2) by 2030 to address health inequities.

## Background

Biomass fuel is solid fuel found in wood plants, crop residues, and animals' dung used for cooking and heating [1]. According to UNSD, World Bank, and WHO, a 2020 joint report,3 billion people who used solid fuel were found in low and middle-income countries [2]. Even since the early 1990s, innovative epidemiological studies showed the interconnected experience of indoor air pollution from solid fuel use with public health consequences [3], developing countries still depend on solid fuel use. Especially in sub-Saharan Africa where households are left with lower options for clean fuel use than to use solid fuels for cooking [4]. The consumption of wood fuel per capita in this region is 2–3 times more likely than in any other region of the world [5]. In Ethiopia, like other countries in Sub-Saharan Africa, traditional cooking technologies with their associated fuels are central practice, which leads to high emissions and health effects [6]. In the region, the use of wood as an energy source accounts for more than 70% [7]. The progress in reducing solid fuel use in the SSA region was from 90% in 1990 to 84% in 2020 [8] which is very slow.

Such Solid fuel use for cooking is the main cause of indoor air pollution in these developing countries [9]. The burning of solid fuels emits a variety of pollutants such as $PM2.5$, $PM10$, $CO$, $NO_2$, organic compounds, and other substances which a had wide range of public health problems [10]. High levels of indoor pollutants are of particular concern since people may spend as much as 90% of their time indoors [11, 12].

If the above-listed solid fuel end and partial products are breathed into the body, acute lower respiratory tract infection, pneumonia, chronic bronchitis, chronic obstructive pulmonary disease, and lung cancer would increase [13, 14]. According to the study report by Global Burden of Diseases (GBD) in 2018, the use of solid fuels for cooking attributed to nearly 1·6 million deaths worldwide [15]. The other evidence from the WHO report, indoor air pollution was attributed to around 3.2 million deaths each year in 2020, and more than 237 000 deaths of children under the age of five [16]. In 2019, 14% of all mortality among children under the

age of 5 in Africa was related to air pollution, which is the third largest predictor for those deaths next to malnutrition, unsafe water, sanitation, and hygiene in SSA [17].

The 2030 agenda for sustainable development also stands for universal access to clean fuel technologies for cooking (SDG 7.1.2) and to reduce substantially the number of deaths due to the joint problems of outdoor and indoor air pollution (SDG 3.9) [18]. Revealing the local potential factors affecting public health including indoor air pollution due to solid fuel use enables the global performance progress in the field. There are different factors affecting the choice of fuel for cooking like housing type, number of rooms in the house, household income, household head sex and age, educational attainment, marital status [19, 20] separate rooms used as a kitchen, media exposure, and residence type. Therefore, this study aimed to investigate factors affecting solid fuel use and spatial variation in Ethiopia using Mini Demographic and Health Surveys 2019 dataset. Thus, open-handed the current evidence into the reality for actors working on solid fuels use to health effects.

## Methods

### Study area and data source

The study was conducted in Ethiopia, which is the second-largest population in Africa. MEDH survey was conducted in nine geographical regions (Tigray, Afar, Amhara, Oromia, Somali, Benishangul-Gumuz, Southern Nations Nationalities and Peoples Region (SNNPR), Gambella, and Harari), and two administrative cities (Addis Ababa and Dire-Dawa) of the country. Like other countries in the world, in Ethiopia, there is Demographic and Health Surveys every five years. We used the fifth Ethiopian Demographic and Health Surveys 2019 (EDHS-2019) database survey in this study. Therefore, this MEDHS is a nationally representative population-based survey with large sample sizes. EDHS data is open source and can be retrieved on the DHS website (https://dhsprogram.com/Data/terms-of-use.cfm).

The 2019 EMDHS sample was a two-stage stratified cluster sample, sampling weights were calculated based on sampling probabilities separately for each sampling stage and each cluster. In the first stage, a total of 305 EAs (93 in urban areas and 212 in rural areas) were selected with probability proportional to EA size (based on the 2019 EPHC frame) and with independent selection in each sampling stratum. In the second stage of selection, a fixed number of 30 households per cluster were selected with an equal probability of systematic selection from the newly created household listing. The field practice was conducted in Adama in clusters that were not part of the 2019 EMDHS sample. EPHI investigators, an ICF technical specialist, an advisor, and representatives from other organizations, including CSA, FMoH, the World Bank, and USAID, supported the data collection. Data collection took place over 3 months, from March 21, 2019, to June 28, 2019. Since this study utilized secondary data, authors had not access direct information from study participants during or after data collection. A total of 9,150 households were selected for the sample, of which 8,794 were engaged. Among involved households, 8,663 were successfully interviewed at a response rate of 99%. Weighted by sampling weight was done to do a reliable statistical analysis.

### Study variables

#### Outcome variables

The dependent variable was fuel type used for a different purpose at the household level. Cooking fuels such as wood, charcoal, straw/shrubs/grass, animal dung, kerosene, and crop residue/wood were categorized as 'solid fuels', while natural gas, liquefied petroleum gas (LPG), biogas, and electricity were classified as 'clean fuels [21]. Then fuel types were categorized as '1'

representing 'solid fuel' and '0' in place of 'clean fuel' to make it convenient for analysis and reporting.

## Predictor variables

**Individual level variables.** Sex of household head (male or female), wealth index (poor, middle, and rich), educational status, having television and radio, a separate room used as a kitchen, and the number of rooms of the household were individual predictor variables.

**Community level variables.** Community level education (lower/higher), place of residence (urban/rural), community level media exposure (exposed/unexposed), region ((recoded as pastoralist region (Benishangul-Gumuz, Somali, Gambella, and Afar), Semi-pastoralist (Oromia, SNNPR), Agrarian (Amhara and Tigray) and City administration (Addis Ababa, and Dire Dawa Harari).

**Data management and analysis.** For data quality assurance purposes, pretests containing in-class training, biomarker training, and field exercise were done. The field exercise was conducted in clusters around Bishoftu which were not included in the 2019 EMDHS sample. A debriefing session was held with the pretest field staff, and adjustments to the questionnaires were done based on lessons drawn from the field practice. The detail of the method part is found in EMDHS 2019 report [22]. Since the data were collected in such organized way, the potential effect of missing data on the outcome variable were insignificant. Therefore, the missing data were excluded in analysis.

Since the outcome variables were dichotomous (solid and clean fuel), bivariable, and multivariable multilevel binary logistic, regression was conducted to assess associations of outcome variables and predictor variables. Independent individual variables in bivariable analysis with p value less than 0.2 were included in multivariable multilevel logistic regression analysis. Then 95% confidence interval (CI) was employed and a p-value less than 0.05 was used for testing statistical significance in multivariable logistic regression analysis [23].

Four models were established; a null model (model 0) without any predictor variable to evaluate the variance of solid fuel use among communities, model 1 included dependent and individual-level predictors, model 2 incorporated dependent and community-level predictors, and model 3 involved the dependent variables and all individual- and community- level predictors. Random effects were measured using cluster variance ($Vc$), a proportional change in variance (($PCV\ ((Vc-Vn)/Vc)$) the intraclass correlation coefficient ($ICC(Vc/(Vc+3.29))$), and the median odds ratio ($MOR\ (\exp\ [(0.95)\sqrt{Vc}])$) [24]. The goodness-of-fit for all models was evaluated using Akaike's information criterion (AIC), Bayesian information criterion (BIC), and Deviance. Then the model with the lower values of all information criteria and deviance were demonstrating the best-fit model [25]. As well as multicollinearity, the effect of independent variables was measured using the variance inflation factor (VIF).

**Spatial autocorrelation.** To explore the geographical distribution, both global and local indicators of spatial correlation are the best imperative tools for solid fuel use within the specified period.

**Global autocorrelations.** To proceed with geographical variation identification of solid fuel use, Global autocorrelations analysis was done. Global Moran's I index was used to detect whether the difference is due to the clustering effect or non-random/dispersion [26]. As well as the simple exploratory spatial analysis was performed to identify the presence of solid fuel use geographical dependence distribution in the country.

**Local statistical analysis.** Further investigation using figures and maps were needed since Global autocorrelations indicate a clustering effect (positive spatial autocorrelation) on solid fuel use over the country. Therefore, hotspot analysis (Gettis-Ord Gi*) was performed to

identify patterns of spatial variation [27] and emphasize the previously stated using global autocorrelations (cluster effect) on solid fuel use. Cluster and outlier analysis (Anselin local Moran's I) was used to describe the spatial patterns of the dependent variables (solid fuel use). This cluster and outlier analysis was used to confirm and accompaniment to show extreme (the hotspot and cold spot) since it permits the identification of groupings and areas where the differences happen [28].

**Spatial scan statistical analysis.** Kuldorff's SaTScan software v.10.1 was used in spatial scan statistical analysis to find the most likely clusters (primary cluster) and another secondary cluster (secondary cluster) [29]. A spatial cluster size < 50%, which is the default of the population, was used as a higher boundary. The primary and secondary clusters were identified and assigned p-values and ranked based on their log-likelihood ratio. That means Spatial scan statistical analysis able to show parts of the country with high solid fuel use using high LLR and p-value to associate with clusters outside of the window.in doing this purely spatial analysis scanning for clusters with high rates Bernoulli model was used. All the above analyses were conducted using Excel Microsoft 2016, STATA 14, ArcGIS 10.7, and SaTScan 10.1 software.

## Ethics approval and consent to participate

The data was obtained online at 'https://www.dhsprogram.com/data/dataset_admin/login_main.cfm.' after we gotten permission from DHS data archivist. However, the EDHS-2019 data collection was reviewed and approved by the Federal Democratic Republic of Ethiopia, Ministry of Science and Technology, and the Institutional Review Board of ICF International. Therefore, for this study, ethics approval and consent to participate is not applicable.

## Results

### Descriptive statistics on socio-demographic characteristics

This study included 8,663 participants. The majority (72.62%) of household heads were males. Approximately seventy percent (69.47%) of participants lived in rural areas. The highest proportion (47.65%) of educational status was no education while the lowest (9.89%) participants were higher in educational status. More than three-quarters (77.10%) of the included study subjects had no television. Most (83.86%) households had no separate room used as the kitchen (Table 1).

### The proportion of solid fuel use

Figs 1 and 2 showed that different types of fuel used in Ethiopia using the EMDHS-2019 dataset. Amongst the solid fuel categories, wood was the highest proportion (71.22%) (Fig 1). Then these fuel types were categorized into solid fuel and clean. The result revealed that the overall national level of solid fuel use was 87.13% (95% CI = (86.4%-87.82%)) (Fig 2). From this 87.13% of solid fuel use, 18.60% accounts for urban and 68.53% rural parts of the country.

### Multilevel logistic regression analysis to determine factors associated with solid fuel use

Educational status, having television, accessing electricity, and wealth index were individual-level variables and community-level education, type of residence, and region were community-level variables significantly associated factors towards solid fuel use in our study.

The odds of solid fuel use among the households not having were 2.56 (AOR = 2.56; 95% CI (1.94–3.38)) times more likely compared to households having a television.

**Table 1. Socio-demographic characteristics of study participants included in the 2019 EMDHS (N = 8,663).**

| Variables | Categories | Frequency (%) |
|---|---|---|
| **Individual Level Variables** | | |
| Age of household head | <30 | 2,520(29.09) |
| | 30–40 | 2,287(26.40) |
| | 41–54 | 1,717(19.82) |
| | >54 | 2,139(24.69) |
| | | The mean age = was 43.05±0.18 |
| Sex of household | Male | 6,291(72.62) |
| | Female | 2,372(27.38) |
| Educational status | No education | 4,128 (47.65) |
| | Primary | 2,715(31.34) |
| | Second | 963(11.12) |
| | Higher | 857(9.89) |
| Wealth index | Poor | 3,498(40.38) |
| | Middle | 1,285(14.83) |
| | Rich | 3,880(44.79) |
| Household with a separate room used as a kitchen | No | 2,400 (83.86) |
| | yes | 462 (16.14) |
| Having radio | No | 6,170 (71.22) |
| | Yes | 2,493(28.78) |
| Number of rooms of the HH | 1 | 6,338 (73.16) |
| | ≥ 2 | 2,325 (26.84) |
| Having Television | No | 6,679 (77.10) |
| | Yes | 1,984(22.90) |
| Community-level media exposure | Unexposed | 5,195(59.97) |
| | Exposed | 3,468(40.03) |
| Community-level educational status | Lower | 4,308(49.73) |
| | Higher | 4,355(50.27) |
| Community level poverty | Higher | 4,276(49.36) |
| | Lower | 4,387(50.64) |
| Residence | Urban | 2,645(30.53) |
| | Rural | 6,018(69.47) |
| Region | Tigray | 714(8.24 |
| | Afar | 664 (7.66) |
| | Amhara | 1,007(11.62) |
| | Oromia | 1,018(11.75) |
| | Somali | 657(7.58) |
| | Benishangul-Gumuz | 734(8.47) |
| | SNNPR | 1,017(11.74) |
| | Gambella | 693(8.00) |
| | Harari | 719(8.30) |
| | Addis Ababa | 702(8.10) |
| | Dire Dawa | 738(8.52) |

The likelihood of using solid fuel was 5.33 (OR = 5.33; 95%CI (3.73–7.63)), 2.92 (OR = 2.92; 95%CI (2.08–4.12)), and 1.53 (OR = 1.53;95%CI (1.12–2.09)), times the more likely educational status of no education, primary education, and secondary education, respectively compared to participants with higher educational status.

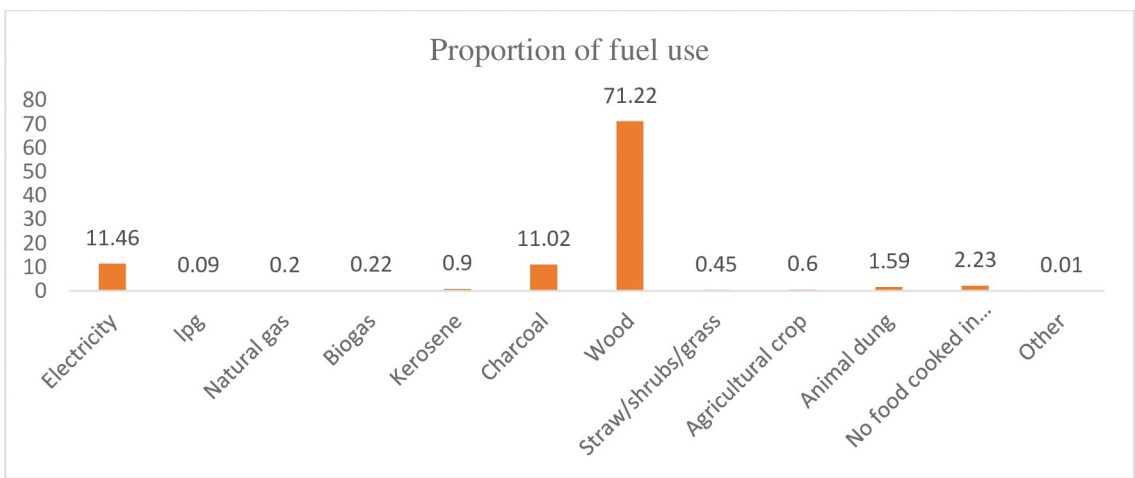

**Fig 1. The proportion of fuel types used in Ethiopia based on the EMDHS 2019 datasets.**

The chance of solid fuel use among the poor was 5.70 (AOR = 5.70; 95% CI ((2.70–19.79))) times higher than the rich. The probability of solid fuel use among the participants without electricity was 9.18 (AOR = 9.18; 95% CI (3.66–23.03)) times more likely as compared with households accessing electricity.

The odds of solid fuel use of the participants whose household had a separate room used as a kitchen were 3.46 (AOR = 3.46; 95% CI (4.35–6.04)) times more likely in solid fuel use than the participants whose household had no kitchen room.

Participants who lived in a rural area had 3.16 (AOR = 3.16; 95% CI ((1.50–6.67)) times chance of solid fuel as compared to participants who lived in the urban area.

The odds of solid fuel use were 5.83 (OR = 5.83; 95% CI (2.24–15.19)) and 8.82 (OR = 8.82; 95% CI (3.75–20.73)) times more likely among the study participants who lived in Semi-pastoralist and pastoralist, respectively compared to the counterparts lived in city administration.

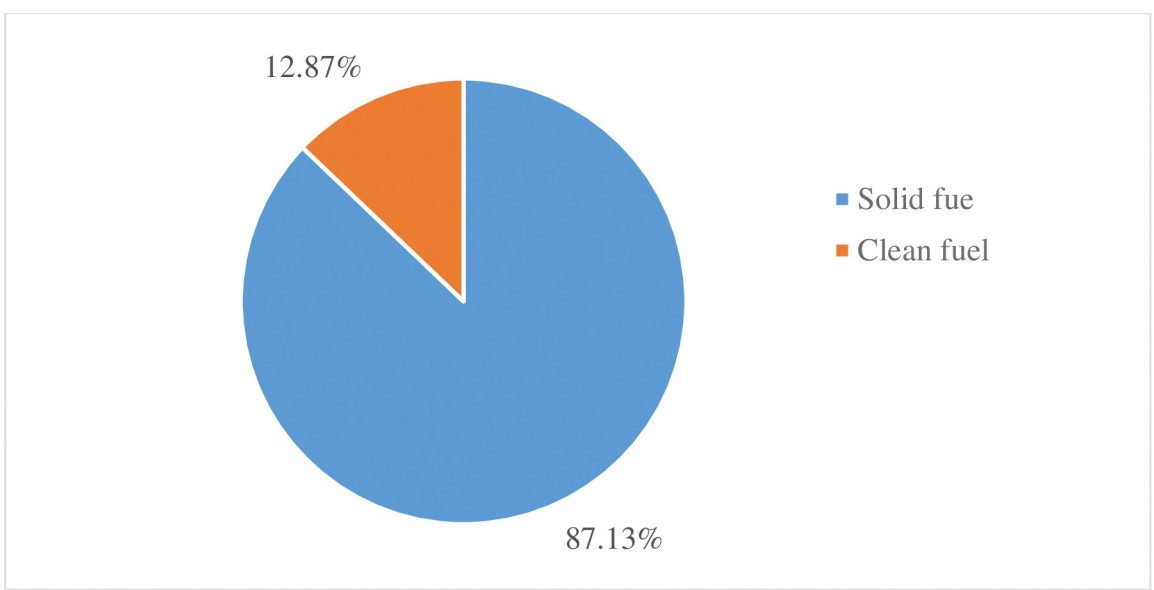

**Fig 2. The proportion of solid and clean fuel types used in Ethiopia based on the EMDHS 2019 datasets.**

The other predictor variable was community-level educational status. Participants with lower level education were 2.76 (OR = 2.76: 95% CI; (1.40–5.430)) times more likely to solid fuel use compared to study subjects with higher-level education (Table 2).

Since there is a significant clustering effect, a Measure of variation (ICC, MOR, and PCV) was performed. The ICC for the total solid fuel use was 0.200, which indicates that the disparity in solid fuel use could be due to unmeasured community-level characteristics. The final model showed that 43.22% of the variance in solid fuel use was explained by both individual and community-level variables. In the empty model indicated MOR was 2.36, the solid fuel use difference between clusters increased by 2.36 times if the individual randomly picked and moves to the next neighbor cluster where the chance of solid fuel use is more prevalent. The final model was the best fit since the goodness of fit test statically values (AIC, BIC, Deviance) were lower than the remaining models (Table 2).

## Spatial pattern of solid fuel use

The spatial distribution of solid fuel use spatially clustered than random chances (z-score = 33.109144, P-value <0.0001) in Ethiopia. 33.109144 of the z-score of Global Moran's analysis revealed that there was less than a 1% chance that this high-clustered pattern was the result due to Random chance. The two extreme tails (red and blue colors) showed that the presence clustered and was significant (Fig 3).

## Hot spot analysis of solid fuel use

Hot spot analysis revealed that the proportion of solid fuel use was highly practiced in most parts of SNNPR, Gambella, Benishangul-Gumuz, and some parts of the Amhara region and Oromia regions. Whereas Addis Ababa, Dire Dawa, and Harari were parts of the country where solid fuel use is lower (Fig 4).

## Cluster and outlier spatial analysis

The map below shows extreme (LL, HH) areas in solid fuel use in the region of Ethiopia. The green color represents "HH" indicating that Gambelia, Benishangul-Gumuz, SNNPR, some parts of Amhara, Oromia, and Afar are parts of the country where solid fuel use is highly prevalent. On the contrary, the red line represents "LL" (Low-Low), indicating that Dire Dawa, some parts of Harari, and Addis Ababa Ethiopia are clustered with lower solid fuel use (Fig 5).

## Kriging interpolation analysis

In the spatial kriging interpolation, analysis the red color showed that solid fuel use was higher in most parts of Ethiopia. On the other hand, the green color indicates the areas (Addis Ababa and some parts of Dire Dawa) use solid fuel to a lower extent (Fig 6).

## Spatial scan statistical analysis

SaTScan spatial analysis of Solid fuel use in Ethiopia generates six windows clusters. The primary cluster window of SaTScan spatial analysis (LLR = 412.59, p<0.000) includes most of SNNPR, Gambella, Benishangul-Gumuz, and some parts of Amhara, and Oromia regions (Table 3 & Fig 7).

## Discussion

This study aimed at investigating the prevalence, predictor variables, and geographical variation of solid fuel use in Ethiopia using the EMDHS 2019 dataset. The current finding revealed

**Table 2. Multilevel regression analysis of solid fuel uses predictors in Ethiopia, EMDHS 2019.**

| Variables | Model 0 | Model 1 AOR (95% CI) | Model 2 AOR (95% CI) | Model 3 AOR (95% CI) |
|---|---|---|---|---|
| **Individual level Factors** | | | | |
| Sex of HHH | | | | |
| Female | | 1.26(1.00,1.59)* | | 1.17(0.93,1.47) |
| Male | | 1 | | 1 |
| Age of HHH | | | | |
| <30 | | 0.89(0.68,1.17) | | 0.96(0.73,1.26) |
| 30–40 | | 0.94(0.70,1.28) | | 1.04(0.77,1.40) |
| 41–54 | | 1.20(0.85,1.70) | | 1.28(0.91,1.81) |
| >54 | | 1 | | 1 |
| Educational status | | | | |
| No education | | 5.65(3.94,8.09)* | | 5.33(3.73,7.63)** |
| Primary | | 3.18(2.25,4.49)** | | 2.92(2.08,4.12)** |
| Secondary | | 1.58(1.16,2.15)* | | 1.53(1.12,2.09)* |
| Higher | | 1 | | 1 |
| Has the HH separate room been used as a kitchen | | | | |
| No | | 5.87(4.73,7.29) ** | | 3.46(4.35,6.04)** |
| Yes | | 1 | | 1 |
| Television | | | | |
| Yes | | 1 | | 1 |
| No | | 3.90(2.92,5.22)** | | 2.56(1.94,3.38)** |
| Has the HH electricity | | | | |
| Yes | | 1 | | 1 |
| No | | 3.01(1.55,5.81)** | | 9.18(3.66,23.03)** |
| Wealth index | | | | |
| Poor | | 6.96(4.36,13.28)** | | 5.70(2.70,19.79)* |
| Middle | | 8.38(6.26,16.96)** | | 1.48(0.09,25.19) |
| Rich | | 1 | | 1 |
| **Community level factors** | | | | |
| Community level education | | | | |
| Higher | | | 1 | 1 |
| lower | | | 4.09(1.81,9.26)* | 2.76(1.40,5.430)* |
| Type of residence | | | | |
| Urban | | | 1 | 1 |
| Rural | | | 4.48(1.46,8.16)** | 3.16(1.50,6.67)* |
| Community-level media exposure | | | | |
| Exposed | | | 1 | 1 |
| Unexposed | | | 2.09(0.77,5.72) | 0.61(0.23,1.60) |
| Region | | | | |
| City administration | | | 1 | 1 |
| Agrarian | | | 1.03(0.31,3.42) | 0.93(0.31,2.80) |
| Semi-pastoralist | | | 9.52(3.28,27.58)** | 5.83(2.24,15.19)** |
| Pastoralist | | | 11.15(8.71,61.54)** | 8.82(3.75,20.73)** |
| VIF | | 3.5 | 2.7 | 2.3 |
| **Measures of variation** | | | | |
| Variance | 0.820 | 0.623 | 0.428 | 0.243 |
| ICC | 0.200 | 0.160 | 0.115 | 0.069 |

(*Continued*)

**Table 2.** (Continued)

| Variables | Model 0 | Model 1 AOR (95% CI) | Model 2 AOR (95% CI) | Model 3 AOR (95% CI) |
|---|---|---|---|---|
| MOR | 2.36 | 2.11 | 1.86 | 1.60 |
| PCV (%) | Ref. | 24.02% | 31.30% | 43.22% |
| **Model fit statistics** | | | | |
| AIC | 6653.864 | 2996.391 | 2854.316 | **2688.114** |
| BIC | 6660.931 | 3052.925 | 2946.185 | **2829.45** |
| LLR | -1370.710 | -1414.158 | -1490.195 | **-1324.057** |
| Deviance | 2,741.420 | 2,828.316 | 2,980.390 | **2,648.114** |

AIC = Akaike's information criterion; BIC = Bayesian information criterion; ICC = intra-class correlation coefficient; LLR = Log Likelihood Ratio; PCV = variance partition coefficients; VIF = variance inflation factor

1 = reference

**P-value < 0.001

*P-value < 0.05, HHH = household head, HH = household, Model 0 (Null model) was fitted without determinant variables. Model 1 is adjusted for individual-level variables. Model 2 is adjusted for community-level variables; Model 3 is the final model adjusted for both individual- and community-level variables.

that the magnitude of solid fuel use was 87.13% (95% CI (86.4%-87.82%)). The overall prevalence solid fuel use of this finding is higher than from the study done in Indian (76%) [30], Myanmar (79.0%) [31], India (72%) [32], and China (54%) [33]. Whereas this value is lower than the findings of the studies conducted in South Africa (90%) [34] and Nepal (88%) [35] of

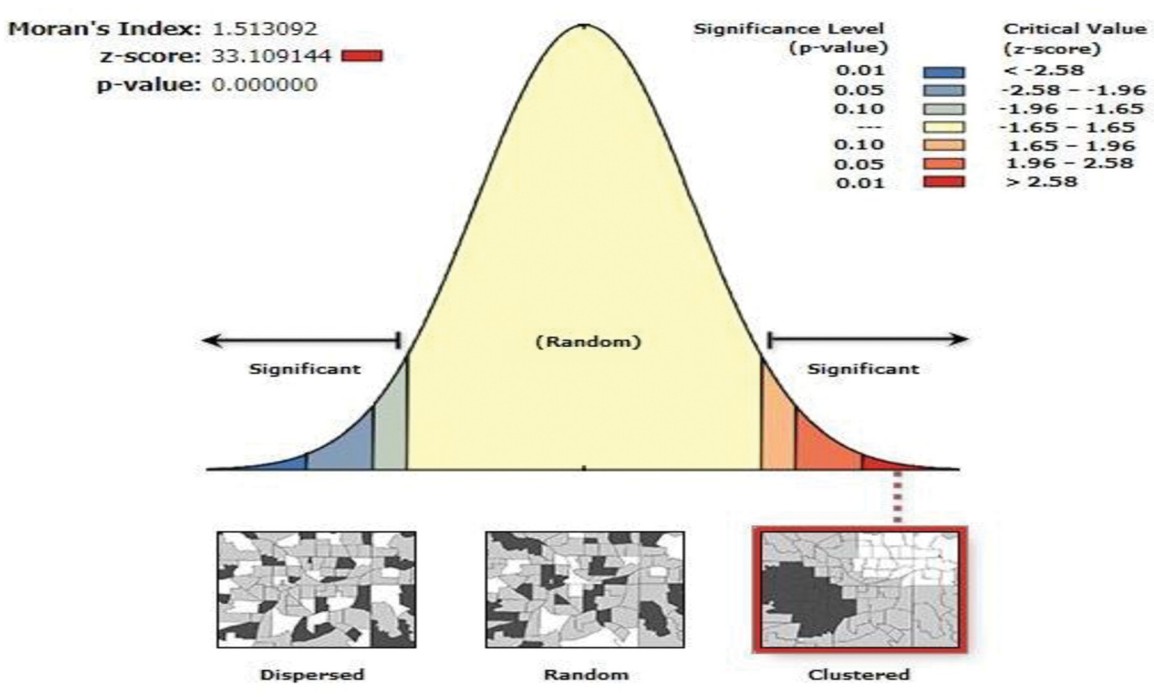

**Fig 3. Spatial patterns of solid fuel use in Ethiopia using the EMDHS 2019 dataset.**

## Hot spot analysis of solid fuel use in Ethiopia, EMDHS 2019

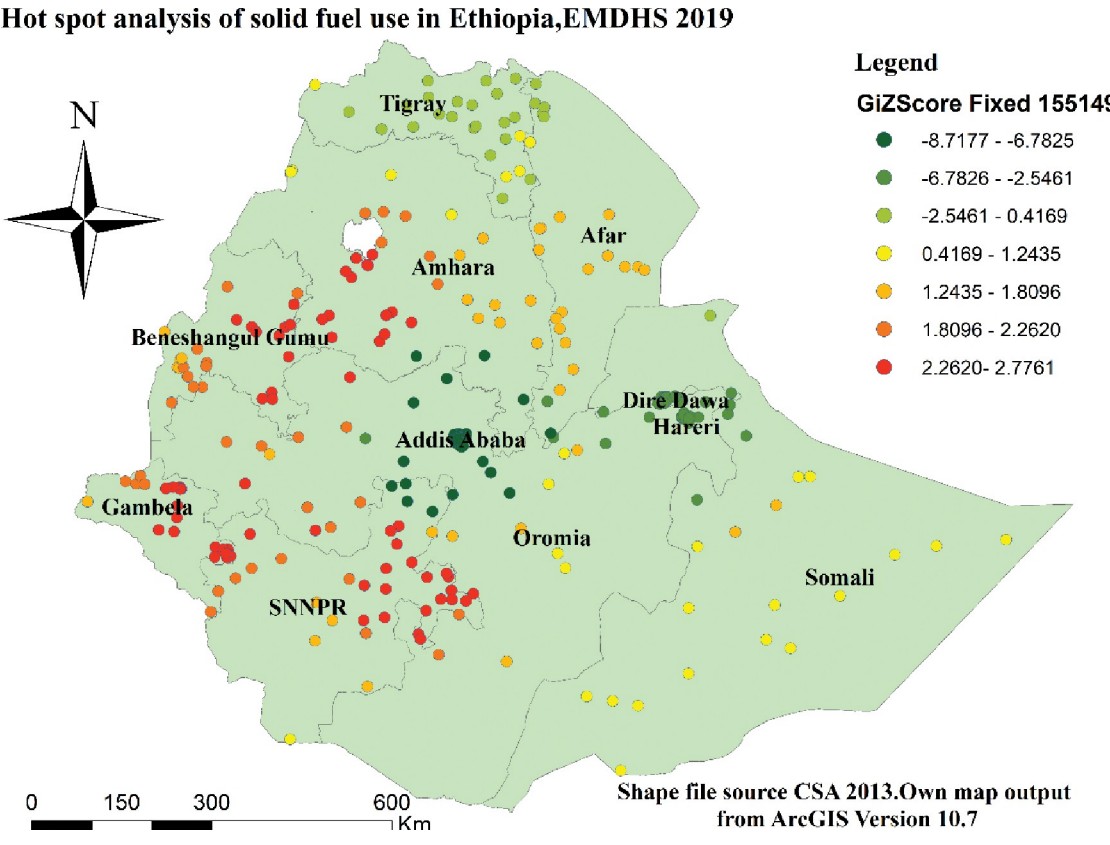

**Fig 4. Hot analysis of solid fuel use in Ethiopia using the EMDHS 2019 dataset.**

the households used solid fuel for domestic purposes. The possible explanation for this variation could be due to sociodemographics variability, economical difference, government concern and the level of development of the countries.

Educational status of the household head, having television, accessing electricity, and wealth index were individual-level variables and community-level education, type of residence, and region were community-level variables significantly associated factors towards solid fuel use in our study.

The practice of solid fuel use of the participants whose households had a separate room used as a kitchen was higher than the participants whose households had no kitchen room. The reason might be due to even if the health effect of solid fuel use is known, having a separate room for the kitchen is the solution for indoor air pollution. Studies showed that houses lacking a separate kitchen room have a greater level of concentrations of particulate matter which leads to exposure to indoor air pollution for young children who spend lots of hours inside the household [36, 37].

The probability of solid fuel use among households not accessing electricity was highly prevalent compared with households accessing electricity. Using electricity as cooking fuel is less time-consuming, less in cost, and a cleaner energy source. Consequently, these and other nature of electricity make it preferable for use than other dirty energy sources.

Type of residence was the other influencing factor affecting the choice of fuel used for cooking. The households found in the rural area relied on solid fuel use for domestic purposes. This might be due to the fact, easy accessibility of dirty/solid fuels like wood plants, animal dung,

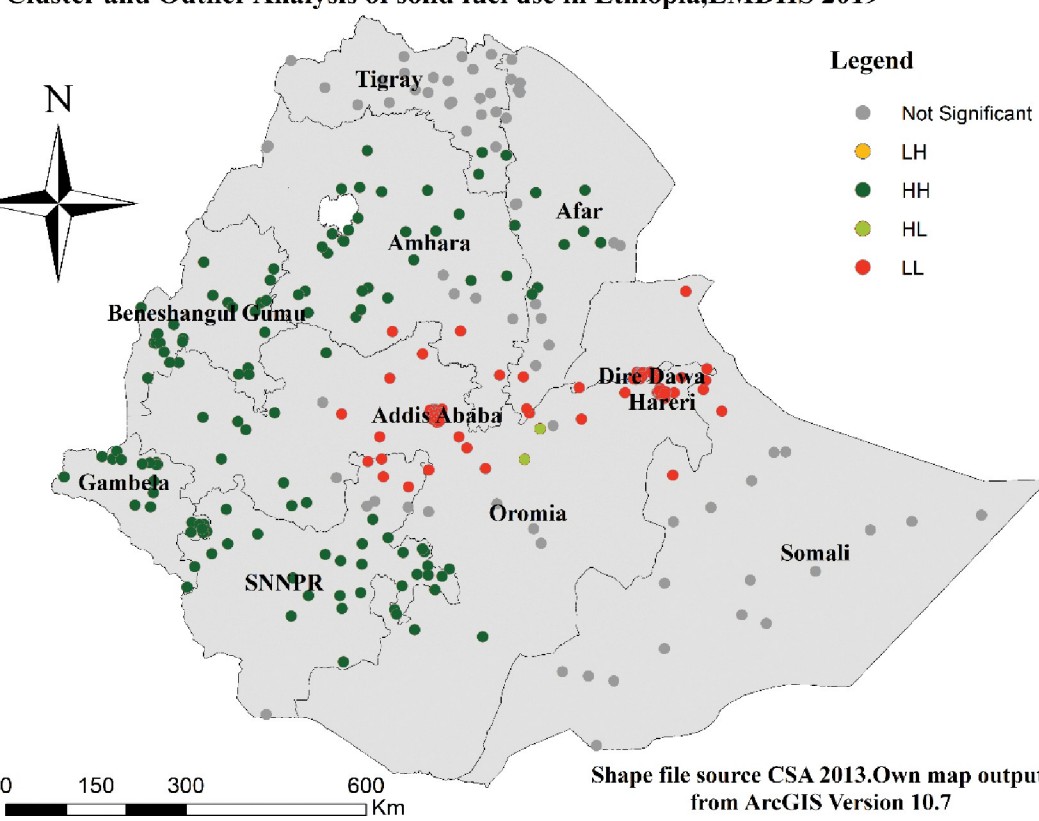

**Fig 5. Cluster and outlier analysis of solid fuel use in Ethiopia using the EMDHS 2019 dataset.**

and agricultural crop products in rural households compared to households in urban areas. This association is supported by a previous study that indicated that rural households in unindustrialized nations still depend on unprocessed solid fuels in the form of wood, dung, and crop residues [38].

The educational status of the household head at the individual and community level was the significantly associated factor with cooking fuel choice. This could be due to educated participants might be informed of the health effect of solid fuel use. The other possible reason, is a higher schooling level, corresponding to enough wages and social status, which may encourage and enable families to select expensive but cleaner fuel as the energy source for cooking [39]. This finding was supported by other previous studies, which indicated that increases in educational level increase the chance of consuming cleaner energy source [40, 41].

Wealth index has also statistically considerable influences on choosing fuel used for cooking. Rich might expend their money on safe life including using clean fuel like electricity, while the poor are struggling in satisfying the daily need of the family. Therefore, their choice could be solid fuel with no or low cost. The energy ladder concept states that as the household income increase, there is an energy transition choice from dirty fuel use to clean /modern energy ladder [42, 43].

This finding confirmed the existence of significant spatial clusters and hot and cold spots in solid fuel use with the help of GIS and Sat Scan statistical analysis overall the country. The first

### Kriging analysis of solid fuel use in Ethiopia,EMDHS 2019

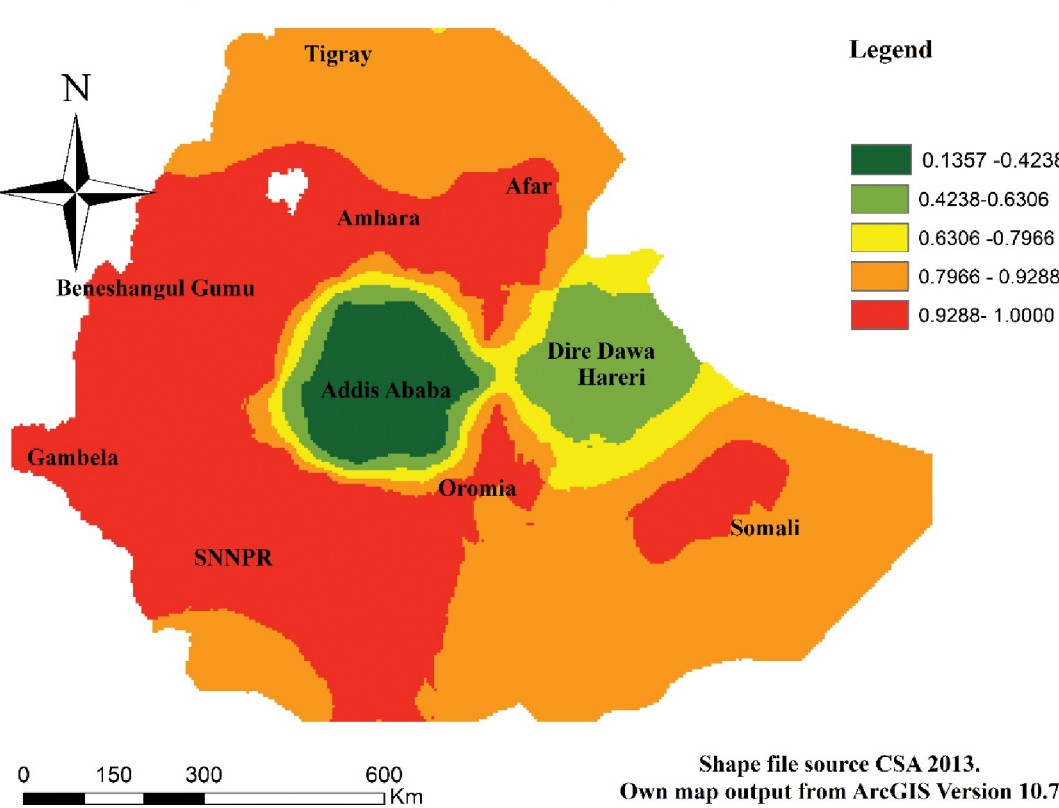

**Fig 6. Kriging analysis of solid fuel use in Ethiopia using the EMDHS 2019 dataset.**

window in cluster analysis included most and different regions of the country whereas the last cluster include Addis Ababa. According to this result, therefore, solid fuel use was significantly higher in SNNPR, Gambella, Benishangul-Gumuz, and some parts of Amhara, and Oromia regions, but it is lower in Addis Ababa. This evidence help to indicate the presences of greater disparity of public health problems related with solid fuel use respiratory problems in Ethiopia for health related policy makers, NGOs and for researchers.

## Conclusion

Based on this finding, the prevalence of solid fuel use was higher in Ethiopia. Educational status, having television, accessing electricity, and wealth index were individual-level variables and community-level education, type of residence, and region were community-level variables statistically significant factors in determining fuel choice for cooking. There was significant spatial variation in solid-fuel use across the country. In order to addressing such heavily dependent on solid fuel use, responsible bodies like health policy makers, national and international organizations, and public health researchers should work on showing health problems of solid fuel use and the means of increasing clean fuel use. Substantial policy modifications are desirable to reach access to clean fuels and technologies (SDG 7.1.2) by 2030 to address health inequities.

**Table 3. Spatial scan analysis of areas with significant solid fuel use in Ethiopia (EMDHS 2019).**

| Clust. No. | Location IDs | Regions | Coordinate/ Radius in km | Popn. | Case | RR | LLR | P-value |
|---|---|---|---|---|---|---|---|---|
| 1 | 219,220,217,218,229, 206,230,211,212, 213, 214,209,207,208,170, 225,228,226,227,221, 224,118,222,210,223,155,94, 215,154, 147, 86,152,153, 194, 216, 200,151,157,201,156, 92,150,146, 120, 149, 195,169,168,167,97,93,96,160,161,158,91,164,196,166,159,148,192,98,163,95,173,204,87,162,191,119,77, 165,80,198,174,179,189,199,190,177,180,193,112,171,176,197,178,52,202,99,72,205,203,75,53,184,76,115,172,70,73,188,74,175,54,187,81,116, 182, 185, 71 | SNNPR, Gambella, Benishangul-Gumuz, Amhara, Oromia | (8.053039N, 33.198166E)/ 609.44 | 3252 | 3212 | 1.23 | 412.59 | <0.0001 |
| 2 | 195,201,194, 96, 200, 223,210,222,224,221, 227,196, 97, 91,226, 228,225,204,215,173, 94, 191, 216, 192, 95, 198,189,120,190,214, 213,179,199, 92, 180, 212,197,177,208,209,93,207,206,211,118, 178,174,230,176,171, 184,87,98,203,115,202,229,112,172,205,168, 188, 169, 218, 217, 182,187,167,185,116, 181,220,186,193,175, 89, 113, 183, 86, 170, 155,119,117,219,164, 99, 154, 156, 274,150 | Amhara, Tigray, Afar, Somali | (7.196976N, 36.099377E)/ 347.76 | 2584 | 2559 | 1.21 | 325.91 | <0.0001 |
| 3 | 142,141,136,125,138, 143,137,123,144,134, 145,111,135,133,110, 114,131,103,122,117, 132,183,102,113,140, 106,129,186,88,89,181,250,105,248,104, 249 | Amhara, Tigray, Afar | (5.479641N, 42.196835E)/ 422.54 | 999 | 995 | 1.16 | 127.94 | <0.0001 |
| 4 | 137,138,123,135,142, 136,145,134,140,131, 141,122,132,133,124, 125,143,144,129,111, 139,121,250,107,248, 110,249,130,255,247, 254 | Oromia, Dire Dawa, Harari | (5.856584N, 43.726017E)/ 420.07 | 855 | 849 | 1.16 | 99.55 | <0.0001 |
| 5 | 44,46,29,45,64,33,62,18,47,30,63,19,48,34,61,20,24,49,66,65,31,51,78,68, 50,26,60,32,23,58,36,25,5,67,43,17,38,3,83,27,2,14,40,37,71,35,69,42,57,82,70,100,13,11,73,16,56,59,76,39,7,12,54,15,72,9,84,127,10,74,126,52,41,53 | Oromia, Dire Dawa, Harari | (11.818783N, 39.955788E)/ 309.08 | 2084 | 1985 | 1.13 | 96.61 | <0.0001 |
| 6 | 3,30,31,26,47,44,32,34,48,45,29,46,64,49, 50,68,63,43,66,19,62,18,24,51,20,65,40,61,1,26,67,42,36,127,78,69 | Addis Ababa | (11.531514N, 40.697674E)/ 241.59 | 953 | 919 | 1.12 | 54.04 | <0.0001 |

## SaTScan analysis of solid fuel use in Ethiopia,EMDHS 2019

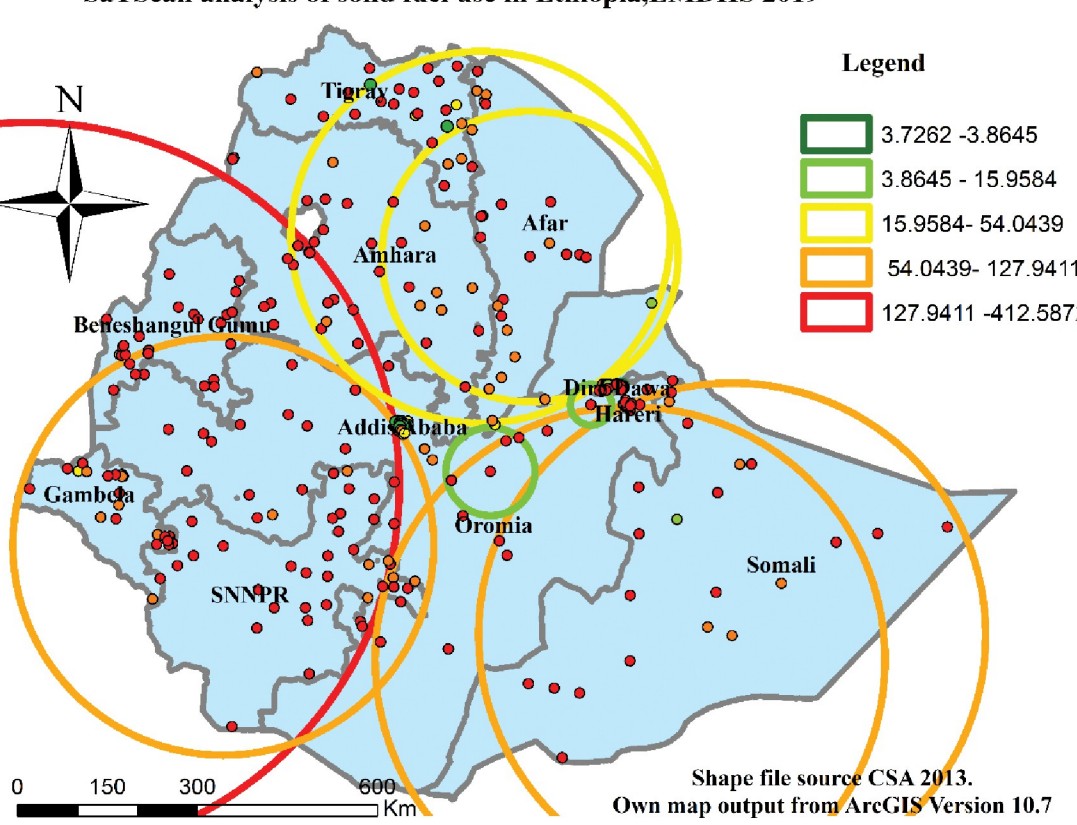

**Fig 7. SaTScan analysis of solid fuel use in Ethiopia EMDHS 2019 dataset.**

## Supporting information

**S1 File.**
(PDF)

## Acknowledgments

Authors would like to thank the Demographic and Health Survey (DHS) Program for permitting us to use the Ethiopian DHS dataset.

## Author Contributions

**Conceptualization:** Jember Azanaw, Gashaw Sisay Chanie.

**Data curation:** Jember Azanaw.

**Formal analysis:** Jember Azanaw.

**Investigation:** Jember Azanaw, Gashaw Sisay Chanie.

**Methodology:** Jember Azanaw.

**Software:** Jember Azanaw.

**Supervision:** Jember Azanaw, Gashaw Sisay Chanie.

**Validation:** Jember Azanaw.

**Visualization:** Jember Azanaw, Gashaw Sisay Chanie.

**Writing – original draft:** Jember Azanaw.

**Writing – review & editing:** Jember Azanaw, Gashaw Sisay Chanie.

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
