## [Decision Letter · Decision Letter 0]

8 Sep 2023

PONE-D-23-14224Spatial variation and determinants of solid fuel use in Ethiopia; Mixed Effect and spatial analysis using 2019 Ethiopia Mini Demographic and Health Survey datasetPLOS ONE

Dear Dr. Azanaw,

Thank you for submitting your manuscript to PLOS ONE. After careful consideration, we feel that it has merit but does not fully meet PLOS ONE’s publication criteria as it currently stands. Therefore, we invite you to submit a revised version of the manuscript that addresses the points raised during the review process. Please submit your revised manuscript by Oct 23 2023 11:59PM. If you will need more time than this to complete your revisions, please reply to this message or contact the journal office at plosone@plos.org. Please include the following items when submitting your revised manuscript:A rebuttal letter that responds to each point raised by the academic editor and reviewer(s). You should upload this letter as a separate file labeled 'Response to Reviewers'.A marked-up copy of your manuscript that highlights changes made to the original version. You should upload this as a separate file labeled 'Revised Manuscript with Track Changes'.An unmarked version of your revised paper without tracked changes. You should upload this as a separate file labeled 'Manuscript'.If applicable, we recommend that you deposit your laboratory protocols in protocols.io to enhance the reproducibility of your results. Protocols.io assigns your protocol its own identifier (DOI) so that it can be cited independently in the future. For instructions see: https://journals.plos.org/plosone/s/submission-guidelines#loc-laboratory-protocols. Additionally, PLOS ONE offers an option for publishing peer-reviewed Lab Protocol articles, which describe protocols hosted on protocols.io. Read more information on sharing protocols at https://plos.org/protocols?utm_medium=editorial-email&utm_source=authorletters&utm_campaign=protocols.

We look forward to receiving your revised manuscript.

Kind regards,

Aiggan Tamene

Academic Editor

PLOS ONE

Journal Requirements:

3. We note that Figures 4 to 7 in your submission contain map images which may be copyrighted. All PLOS content is published under the Creative Commons Attribution License (CC BY 4.0), which means that the manuscript, images, and Supporting Information files will be freely available online, and any third party is permitted to access, download, copy, distribute, and use these materials in any way, even commercially, with proper attribution. For these reasons, we cannot publish previously copyrighted maps or satellite images created using proprietary data, such as Google software (Google Maps, Street View, and Earth). For more information, see our copyright guidelines: http://journals.plos.org/plosone/s/licenses-and-copyright.

(1) You may seek permission from the original copyright holder of Figures 4 to 7 to publish the content specifically under the CC BY 4.0 license.  

Reviewers' comments:

Reviewer's Responses to Questions

**Comments to the Author**

1. Is the manuscript technically sound, and do the data support the conclusions?

Reviewer #1: Yes

Reviewer #2: Yes

2. Has the statistical analysis been performed appropriately and rigorously? 

Reviewer #1: Yes

Reviewer #2: Yes

3. Have the authors made all data underlying the findings in their manuscript fully available?

Reviewer #1: Yes

Reviewer #2: Yes

4. Is the manuscript presented in an intelligible fashion and written in standard English?

Reviewer #1: Yes

Reviewer #2: Yes

5. Review Comments to the Author

Reviewer #1: Quite well written. You may want to consider including information on how missing data was managed, and also elaborating on the public health implications of spatial clusters and outliers for solid fuel use. You may also want to suggest specific policy changes or interventions based on the results.

Reviewer #2: the manuscript addressed a critical topic. Yet I have some comments

1. There exists a discrepancy between the EMDHS and the manuscript results especially magnitude of solid fuel use.

2. the discussion and conclusion are not well articulated

6. PLOS authors have the option to publish the peer review history of their article (what does this mean?). If published, this will include your full peer review and any attached files.

Reviewer #1: **Yes: **Abdul-Basit Abdul-Samed

Reviewer #2: **Yes: **Abel Afework Belayneh

---

## [Author Response · Author response to Decision Letter 0]

4 Nov 2023

Response to Reviewers and Academic Editor

Dear; Editor and Reviewers, Please accept our revised manuscript and note our point-by point response to Academic editor and reviewers below for the manuscript titled “Spatial variation and determinants of solid fuel use in Ethiopia; Mixed Effect and spatial analysis using 2019 Ethiopia Mini Demographic and Health Survey dataset” ,,with Manuscript ID. PONE-D-23-14224”

 Responses for each issue raised by Academic Editor and Reviewers indicated by highlight.

Response to Academic Editor

Authors’ response: Thank you dear editor, we have checked and corrected as per suggested.

2. In your Data Availability statement, you have not specified where the minimal data set underlying the results described in your manuscript can be found. PLOS defines a study's minimal data set as the underlying data used to reach the conclusions drawn in the manuscript and any additional data required to replicate the reported study findings in their entirety. All PLOS journals require that the minimal data set be made fully available. Upon re-submitting your revised manuscript, please upload your study’s minimal underlying data set either as Supporting Information files or to a stable, public repository and include the relevant URLs, DOIs, or accession numbers within your revised cover letter. Any potentially identifying patient information must be fully anonymized.

Important: If there are ethical or legal restrictions to sharing your data publicly, please explain these restrictions in detail. Please see our guidelines for more information on what we consider unacceptable restrictions to publicly sharing data: http://journals.plos.org/plosone/s/data-availability#loc-unacceptable-data-access-restrictions. Note that it is not acceptable for the authors to be the sole named individuals responsible for ensuring data access. We will update your Data Availability statement to reflect the information you provide in your cover letter. 

Authors’ response: Thank you, but presenting the ethical approval, consent for participation and anonymity might not be necessary as well as it is impossible for us since it is secondary data. But there is permission to use this data by Data Archivist of the Demographic and Health Surveys (DHS) Program at data file reference references@dhsprogram.com. Data Availability statement that we used expressed in the manuscript.

3. We note that Figures 4 to 7 in your submission contain map images, which may be copyrighted. All PLOS content is published under the Creative Commons Attribution License (CC BY 4.0), which means that the manuscript, images, and Supporting Information files will be freely available online, and any third party is permitted to access, download, copy, distribute, and use these materials in any way, even commercially, with proper attribution. 

Authors’ response: We are so grateful for this comment, we have included the figure generated for ArcGIS version 10.7.1 and for the data, and we have permission from Data Archivist of the Demographic and Health Surveys, which expressed in the manuscript. Therefore, there is no need of permission regarding included figures from copyrighted.

4. Please review your reference list to ensure that it is complete and correct. If you have cited papers that have been retracted, please include the rationale for doing so in the manuscript text, or remove these references and replace them with relevant current references. Any changes to the reference list should be mentioned in the rebuttal letter that accompanies your revised manuscript. If you need to cite a retracted article, indicate the article has retracted status in the References list and include a citation and full reference for the retraction notice.

Authors’ response: Thank you, we tried to check and revised the references part of the manuscript. 

Reviewer #1 

Quite well written. You may want to consider including information on how missing data was managed, and elaborating on the public health implications of spatial clusters and outliers for solid fuel use. You may also want to suggest specific policy changes or interventions based on the results.

Authors’ response: Thank you very much dear reviewer for your time and effort for showing authors gaps and for constructive comments to be corrected. Since the missing of pertinent data were insignificant effect on the outcome variable, it was not reported, but now we include this concept in the manuscript. Therefore, we revised the manuscript as suggested by the reviewer.

Reviewer #2

The manuscript addressed a critical topic. Yet I have some comments

1. There exists a discrepancy between the EMDHS and the manuscript results especially magnitude of solid fuel use.

Author’s response: We, authors would really thank for pointing us such deep insight. The discrepancy of the result in this study and EMDHS 2019 is the difference in classification of kerosene. In our study it was classified as solid fuel basing on different articles like “Household cooking fuel type and childhood anemia in sub-Saharan Africa: analysis of cross-sectional

Surveys of 123, 186 children from 29 countries” whereas EMDHS 2019 classified it as clean fuel.

2. The discussion and conclusion are not well articulated

Authors’ response: Thank you so much for these valid comments which helped us in improving the discussion and conclusion write-up. We are highly benefited from these comments and tried to rewrite the discussion and conclusion sections.

---

## [Decision Letter · Decision Letter 1]

10 Nov 2023

Spatial variation and determinants of solid fuel use in Ethiopia; Mixed Effect and spatial analysis using 2019 Ethiopia Mini Demographic and Health Survey dataset

PONE-D-23-14224R1

Dear Dr. Azanaw,

We’re pleased to inform you that your manuscript has been judged scientifically suitable for publication and will be formally accepted for publication once it meets all outstanding technical requirements.

Kind regards,

Mohammed Feyisso Shaka, MPH

Academic Editor

PLOS ONE

Additional Editor Comments (optional):

Reviewers' comments:

Reviewer's Responses to Questions

**Comments to the Author**

1. If the authors have adequately addressed your comments raised in a previous round of review and you feel that this manuscript is now acceptable for publication, you may indicate that here to bypass the “Comments to the Author” section, enter your conflict of interest statement in the “Confidential to Editor” section, and submit your "Accept" recommendation.

Reviewer #2: All comments have been addressed

2. Is the manuscript technically sound, and do the data support the conclusions?

Reviewer #2: Yes

3. Has the statistical analysis been performed appropriately and rigorously? 

Reviewer #2: Yes

4. Have the authors made all data underlying the findings in their manuscript fully available?

Reviewer #2: Yes

5. Is the manuscript presented in an intelligible fashion and written in standard English?

Reviewer #2: Yes

6. Review Comments to the Author

Reviewer #2: You have addressed all my concerns as per my comment. But I suggest if you better increase the image quality of figure 3.

7. PLOS authors have the option to publish the peer review history of their article (what does this mean?). If published, this will include your full peer review and any attached files.

Reviewer #2: **Yes: **Abel Afework

---

## [Editor Report · Acceptance letter]

21 Nov 2023

PONE-D-23-14224R1 

Spatial variation and determinants of solid fuel use in Ethiopia; Mixed Effect and spatial analysis using 2019 Ethiopia Mini Demographic and Health Survey dataset 

Dear Dr. Azanaw:

I'm pleased to inform you that your manuscript has been deemed suitable for publication in PLOS ONE. Congratulations! Your manuscript is now with our production department. 

Kind regards, 

on behalf of

Mr. Mohammed Feyisso Shaka 

Academic Editor

PLOS ONE